

# Multi-year statistical and modelling analysis of submicrometer aerosol number size distributions at a rain forest site in Amazonia

Luciana Varanda Rizzo[1], Pontus Roldin[2], Joel Brito[3,*], John Backman[4,**], Erik Swietlicki[2], Radovan Krejci[5], Peter Tunved[5], Tukka Petäjä[4], Markku Kulmala[4], and Paulo Artaxo[3]

[1]Departamento de Ciências Ambientais, Universidade Federal de São Paulo, Diadema, Brazil
[2]Physics Institute, Lund University, Lund, Sweden
[3]Instituto de Física, Universidade de São Paulo, São Paulo, Brazil
[4]Department of Physics, University of Helsinki, Helsinki, Finland
[5]Department of Environmental Science and Analytical Chemistry (ACES), Stockholm University, Stockholm, Sweden
[*]now at: Laboratory for Meteorological Physics, Université Clermont Auvergne, Clermont-Ferrand, France
[**]now at: Finnish Meteorological Institute, Atmospheric Composition Research, Helsinki, Finland

*Correspondence to:* Luciana V. Rizzo (lrizzo@unifesp.br)

**Abstract.** The Amazon Basin is a unique region to study atmospheric aerosols, given their relevance for the regional hydrological cycle and large uncertainty of their sources. Multi-year datasets are crucial when contrasting periods of natural conditions and periods influenced by anthropogenic emissions. In the wet season, biogenic sources and processes prevail, and the Amazonian atmospheric composition resembles pre-industrial conditions. In the dry season, the Basin is influenced by widespread

biomass burning emissions. This work reports multi-year observations of high time resolution submicrometer (10-600 nm) particle number size distributions at a rain forest site in Amazonia (TT34 tower, 60 km NW from Manaus city), between years 2008-2010 and 2012-2014. Median particle number concentration was 403 cm$^{-3}$ in the wet season and 1254 cm$^{-3}$ in the dry season. The Aitken mode ($\sim$30-100 nm in diameter) was prominent during the wet season, while accumulation mode

($\sim$100-600 nm in diameter) dominated the particle size spectra during the dry season. Cluster analysis identified groups of aerosol number size distribution influenced by convective downdrafts, nucleation events and fresh biomass burning emissions. New particle formation and subsequent growth was rarely observed during the 749 days of observations, similar to previous observations in the Amazon Basin. A stationary 1D column model (ADCHEM - Aerosol Dynamics, gas and particle phase CHEMistry and radiative transfer model) was used to assess importance of processes behind the observed diurnal particle size

distribution trends. Three major particle source types are required in the model to reproduce the observations: (i) a surface source of particles in the evening, possibly related to primary biological emissions (ii) entrainment of accumulation mode aerosols in the morning, and (iii) convective downdrafts transporting Aitken mode particles into the boundary layer mostly during the afternoon. The latter process has the largest influence on the modelled particle number size distributions. However, convective downdrafts are often associated with rain and thus act both as a source of Aitken mode particles, and as a sink of ac-

cumulation mode particles, causing a net reduction in the median total particle number concentrations in the surface layer. Our study shows that the combination of the three mentioned particle sources are essential to sustain particle number concentrations in Amazonia.



## 1 Introduction

The Amazon Basin contains some of the continental areas worldwide where aerosol number concentrations as low as 300-500 particles per $cm^{-3}$ are routinely observed (Martin et al., 2010b). This concentration range represents an upper limit to the natural atmospheric particle loading before anthropogenic influence (Andreae, 2007). Yet, even the most preserved forest areas in the region are seasonally influenced by in-basin anthropogenic emissions resulting from expansion of agriculture, logging and urbanization (Davidson et al., 2012), as well as out-of-basin long range transport (Andreae et al., 2015). Characterization of Amazonian aerosols under clean and polluted conditions is crucial part to our understanding of the biosphere-atmosphere dynamical processes and ecosystem resilience to changes (Artaxo et al., 2013). Moreover, the intense tropical convective activity makes the Amazon Basin a relevant source of particles to the global atmosphere (Andreae et al., 2001; Andreae and Crutzen, 1997). In last 20 years, there has been an effort to characterize atmospheric aerosols in Amazonia through intensive studies (Martin et al., 2010b). This work reports long-term observations of high time resolution aerosol properties observations in central Amazonia. This activity was carried out as a part of the projects EUCAARI (European Integrated Project on Aerosol Cloud Climate and Air Quality interactions, Kulmala et al., 2011), AEROCLIMA (Direct and indirect effects of aerosols on climate in Amazonia and Pantanal, Artaxo et al., 2013) and GoAmazon2014/5 (Martin et al., 2016). Special focus in this work is on submicrometer particle number size distributions.

Particle diameter and number concentration are key physical parameters, defining the aerosol impacts on climate and human health. The aerosol impact on climate is driven by its direct and indirect interaction with solar radiation (IPCC, 2014). One of the most fundamental property of aerosol particles is their diameter, which strongly influences their ability to act as cloud condensation nuclei (CCN) (Dusek et al., 2006) and their interaction with radiation (Bohren and Huffman, 2008; McFiggans et al., 2006). Particle diameter also determines the spectral dependence of light scattered by the particles (Schuster et al., 2006). The ability of aerosol particles to penetrate into the human respiratory system is also a function of particle size (Löndahl et al., 2008), and several studies associate exposure to ambient aerosols with increased risk of mortality (e.g., Pope and Dockery, 2006). Moreover, through particle number size distribution analysis, it is possible to investigate how the particle population is influenced by dynamical processes like primary emissions, transport, condensation, new particle formation, coagulation, dry and wet deposition and in cloud processing.

In Amazonia, organic carbon accounts for 60-70% of particle mass in the fine mode (diameter < 2.5 $\mu m$). The remaining fraction contains mainly soil dust particles, sulfate and elemental carbon (Martin et al., 2010b). Relative proportions of major components vary between the wet season, when clean conditions prevail, and the dry season, when the whole Basin is affected by regional biomass burning emissions. The relative contribution of primary and secondary organic carbon particle sources in Amazonia is still under investigation, although observations indicate that secondary organic aerosols (SOA) play a major role for the fine mode particle chemical composition (Chen et al., 2009, 2015; Fuzzi et al., 2007; Pöschl et al., 2010). SOA is formed when volatile organic compounds (VOCs) are oxidized and produce low volatility oxidation products. These products can either condense on pre-existent particles (Pöhlker et al., 2012a) or contribute to new particle formation (NPF) and subsequent growth





(Riipinen et al., 2012). The latter is an important source of particles, as well as CCN, to the global atmosphere (Kerminen et al., 2012; Spracklen et al., 2008, 2010).

NPF events are typically characterized by a fast increase in number concentration of particles with diameters in the range of 1 to 3 nm followed by their growth. Observations have shown that sulfuric acid, amines and organic vapor play a key role in the activation and subsequent growth of particle clusters (Almeida et al., 2013; Ehn et al., 2014; Kulmala et al., 2013; Paasonen et al., 2010). NPF have been observed worldwide and in different environments (Kulmala et al., 2004), but several intensive experiments have shown that in Amazonia nucleation and subsequent particle growth is rare inside the boundary layer (BL) (Rissler et al., 2004; Rizzo et al., 2010; Zhou et al., 2002). Since the occurrence of NPF may depend on season and location (i.e., may vary depending on atmospheric conditions), short duration experiments cannot be conclusive about its frequency in Amazonia. The multi-year observations reported here confirm that NPF is uncommon, at least at this forest site in Central Amazonia.

While NPF within the BL may not be a relevant source of fine mode particles in Amazonia, there are other possible sources to support the observed aerosol loadings during the wet season. Observations of submicrometer primary biological aerosols in Amazonia have been reported, likely consisting of individual bacterial cells, fragments of biological particles and droplets of organic and inorganic solutions ejected by plants and microorganisms (China et al., 2016; Elbert et al., 2007; Huffman et al., 2012; Krejci et al., 2005a; Pöhlker et al., 2012a). Another possible source is the downward transport of particles from above the BL, first suggested by Zhou et al. (2002). Observations reporting high concentrations of nucleation mode particles in the outflow of deep convective clouds reinforce this hypothesis (Krejci et al., 2003). The downward transport of particles produced above the BL can occur either through strong convective downdrafts (Wang et al., 2016), typically observed in Amazonia (Machado et al., 2004; Nunes et al., 2016), and through continuous entrainment. The origin of these particles in the free troposphere (FT) is not understood. Possible processes might involve in cloud processing (e.g., Hoppel et al., 1994; Lee et al., 2004), NPF at high altitudes driven by highly oxygenated organic molecules (Bianchi et al., 2016), or formation through heterogeneous pathways (Jang et al., 2002; Lim et al., 2005; Limbeck et al., 2003). Measurements of submicrometer organic aerosol composition in Amazonia indicated that gas-phase and particle-phase pathways of secondary aerosol formation have comparable strengths in the wet season (Chen et al., 2015).

In this paper, the seasonal and diurnal variability of submicrometer particle number size distributions in a forest site in Amazonia will be presented. The most relevant particle sources, sinks and dynamical processes are investigated using a stationary 1D column model (ADCHEM - Aerosol Dynamics, gas and particle phase CHEMistry and radiative transfer model), as well as statistic tools, especially cluster analysis.

## 2 Methods

### 2.1 Site description and supporting measurements

Particle number size distributions were measured at the Cuieiras forest reservation in Central Amazonia. The reservation is surrounded by primary tropical forest areas within a radius of about 500 km, except for the SE direction, typically downwind,





in which the Manus city sits 60 km away. Measurements were carried out in two periods: from February 2008 to July 2010, and from November 2012 to October 2014. The reserve is located 60 km NW of Manaus, a developing city with 1.8 million inhabitants (IBGE, 2015). No biomass burning takes place in the reservation, but the site is affected by regional transport of biomass burning pollutants, especially in the dry season (July-December). Occasionally, the Manaus urban plume reaches the

site (Rizzo et al., 2013). Most of the time, the prevailing trade winds blow over vast expanses of intact tropical forest before reaching the measurement tower (TT34) (2°35.6570′ S, 60°12.5570′ W, 110m a.s.l.). Aerosol measurements were taken 10 m above the canopy top (39 m above ground) through inlet lines leading to an air conditioned container at ground level where particle analyzers were placed. The laminar-flow inlet had 50% aerodynamic cutoffs of 4 nm and 7 $\mu m$ for our flow conditions. All measurements were taken under low relative humidity (RH) conditions (30–40%), assured by an automatic diffusion dryer

(Tuch et al., 2009). Supporting aerosol measurements included particle absorption and scattering coefficients, respectively using a Multi-Angle Absorption Photometer (MAAP 5012, Thermo Inc.) and a Integrating Nephelometer (Model 3563, TSI Inc.). Housing for the researchers and a 60 kW diesel generator that provided power supply were located 0.33 km and 0.72 km respectively to the west of the sampling site (downwind). A detailed description of the measurement site and surrounding can be found in Martin et al. (2010a).

In this work we define wet season as the period from January to June, and dry season from July to December, in line with Rizzo et al. (2013). This definition may not be strictly correct from a climatological point of view, since the start date of each season changes from one year to another, depending on precipitation patterns. Nevertheless, for the purpose of the current aerosol dataset interpretation, this is a reasonable choice. A detailed description of climatological conditions can be found in Marengo et al. (2001).

Meteorological parameters were measured at the K34 tower (2°36.5450′ S, 60°12.5580′ W, 130 m a.s.l.), located 1.6 km to the south of the aerosol sampling site, and maintained by the National Institute for Research in the Amazon (INPA). In this article, the terminology from Rissler et al. (2006) was adopted to define the BL structure into the sublayers: convective boundary layer (CBL), nocturnal stable layer (NL) and residual layer (RL). Variations on surface layer equivalent potential temperature ($\theta_e$) were used as a proxy to the occurrence of downdrafts, using a method similar to Wang et al. (2016). First,

$\theta_e$ time series were detrended by subtraction of each seasonal mean value. Then, a time series of $\Delta\theta_e$ was determined by subtracting the detrended $\theta_e$ values from its seasonal mean diurnal cycle (Fig. S1). In this way, $\Delta\theta_e$ refers to a deviation from the mean diurnal behavior, so that if $\Delta\theta_e < 0$, it means that the observed $\theta_e$ is below the expected for that time of day and season. Decreases in $\theta_e$ have been associated to convective downdrafts in Amazonia, and to the transport of ozone to the surface (Betts et al., 2002; Gerken et al., 2016).

## 2.2   Particle number size distribution measurements

Three submicrometer particle spectrometer systems were used intermittently at the TT34 forest site. A custom-built Scanning Mobility Particle Sizer (SMPS) system, provided by the Aerosol Group at the Division of Nuclear Physics at Lund University, was operated between February 2008 and July 2010, consisting of a bipolar charger, a medium-long Vienna-type Differential Mobility Analyzer (DMA) and a butanol based Condensation Particle Counter (CPC 3010, TSI Inc.). This system, here referred





to as the Lund SMPS, provided particle number size distributions in the size range from 10 to 600 nm every 5 minutes, and was controlled through a Lab View 7.1 based program.

A TSI SMPS system (classifier model 3080, DMA model 3081 and CPC model 3010) operated in October 2008 and in July to October 2013. The system delivered particle number size distributions in the size range 10-500 nm every 5 minutes. Lund

and TSI SMPS systems were inter-compared between 4–16 October 2008, showing satisfactory linear correlations for total number concentrations (slope = 0.89, $R^2$ = 0.97) and mean geometric diameters (slope = 0.85, $R^2$ = 0.86). Median bin to bin ratios where within ±20% in the size range of 30–300 nm.

A custom-built Differential Mobility Particle Sizer (DMPS) system, provided by the Division of Atmospheric Sciences at Helsinki University, operated in Nov-Dec 2012 and in 2014. The system, operating with a butanol based CPC (TSI Inc., 3772),

provided particle number size distributions in the range of 6 to 800 nm every 10 minutes. High aerosol and sheath flow were used to measure particles in the 6 to 280 nm size range and low flows for 280 to 800 nm sized particles. Particle number size distributions and total number concentrations were interchangeably measured using an automated bypass valve in every measurement cycle to provide a means to validate the DMPS performance.

Ambient temperature and pressure were used to adjust particle number concentrations to standard temperature and pressure

conditions (T =293.15 K, P =1013.25 hPa). The quality of particle number size distribution measurements depends essentially on the stability of sample and sheath air flow rates, as well as on the performance of the CPC (Wiedensohler et al., 2012). Concerning to the CPC performance, butanol based CPC counting efficiency decreases as water vapor condenses inside the saturator. Since this experiment was conducted in a remote area with not easy access, it was not possible to replace butanol in the SMPS systems more frequently than once every 2–3 weeks. For data quality assurance, three CPCs, independent of SMPS

systems, were intermittently operated: TSI-3010, TSI-3772 and TSI-3782 models. The latter, a water based CPC, showed a counting efficiency 20% smaller in comparison to the other models. Particle number concentrations measured by the independent CPCs were compared to integrated particle number concentration from size distributions, with satisfactory linear correlations ($R^2$ ranging between 0.83 and 0.99). Only size distribution samples correspondent to CPC/SMPS particle number concentration ratios within 1.0±0.2 were considered in this analysis. For the periods when CPC measurements were not

available, data quality was assured via Mie scattering closure studies, comparing particle light scattering coefficients calculated from input size distributions against measured scattering coefficients.

Results from instrument inter-comparison workshops indicate that, under controlled laboratory conditions, the particle number size distributions from 20 to 200 nm determined by mobility particle size design are within an uncertainty range of around ±10% after correcting for internal particle losses (Wiedensohler et al., 2012). For particles in the size range of 10-100 nm,

Hornsby and Pryor (2014) report an agreement between several TSI SMPS systems and a TSI FMPS system (Fast Mobility Particle Sizer) within ±30% for particle number concentration and within ±7.5% for geometric mean diameter ($D_{pg}$). Those values can be considered a lower limit for the uncertainties in long-term field measurements, especially at locations with difficult maintenance.

In total, 184040 submicrometer particle number size spectra were measured between February 2008 and October 2014. It

was detected that 2.4% of the samples were contaminated by the power supply diesel generator, and these were removed from





the dataset. Contamination was detected based on fast variations on particle number concentration (>100% in 5 min) associated with northwestern winds (270-360°).

## 2.3 Data processing

Particle size distribution data was interpolated to a common diameter sequence. Changes to the integrated particle number

concentration remained below 0.5%. Particle size distributions were fitted by a multimodal log normal distribution function, described by the following equation:

$$f(D_p, D_{pg_i}, N_i, \sigma_i) = \sum_{i=1}^{n} \frac{N_i}{\sqrt{2\pi}\log(\sigma_i)} \exp\left\{ -\frac{[\log(D_p) - \log(d_i)]^2}{2\log^2(\sigma_i)} \right\} \tag{1}$$

where $D_p$ is the particle diameter. Each of the $n$ lognormal modes is characterized by 3 parameters: the mode number concentration $N_i$, the mode geometric mean diameter $d_i$, and the mode geometric standard deviation $\sigma_i$. A script was developed

to fit one to three lognormal modes to the measured particle number size distributions. The algorithm was based on Hussein et al. (2005) guidelines. Initially, the algorithm decides between two or three modes least squares fit based on the percent concentration of particles with diameters less than 40 nm and of particles with diameters greater than 100 nm. There are constraints for the mean geometric diameter of the first mode (maximum 25 nm) as well as for the geometric standard deviation of all modes (minimum 1.2, maximum 2.1). The algorithm uses previous fitting parameters as a start point for the current fitting

procedure. As a measure of fitting quality, the algorithm uses the root mean square error and a comparison between integrated particle number concentrations calculated from the measured size distribution and from the fitted curve. The algorithm is able to reduce the number of modes if it detects superposition, based on mean geometric diameters and concentration ratios between adjacent modes. It is also able to increase the number of modes if it produces a major improvement on fitting quality. As an indication of the successfulness of the algorithm, the average ratio between the integrated particle number concentration of

measured and lognormal fitted size distributions was 0.99 with $R^2 = 1.0$.

### 2.3.1 Cluster analysis

A clustering method was applied on the particle number size distribution data, in order to group particle size spectra with similar shape and to unveil relationships with other variables such as time of occurrence, season, local wind direction and particle absorption coefficient. For that, particle concentration variability was removed through normalization of each size

spectra with respect to its maximum concentration. Cluster analysis has been applied on particle number size distribution data before (e.g., Beddows et al., 2009; Dall'Osto et al., 2010; Tunved et al., 2004), with *k-means* being the most used clustering technique. Here, the function *kmeans.m* was used, available in MATLAB. It uses a two-phase iterative algorithm to minimize the sum of point-to-centroid distances, summed over all clusters. Since this algorithm is very sensitive to outliers, these were omitted (0.13% of samples). The inner fences ($f_1$ and $f_3$) of the data were calculated using the interquartile range ($Q_3 - Q_1$)

as $f_1 = Q_1 - 1.5(Q_3 - Q_1)$ and $f_3 = Q_3 + 1.5(Q_3 - Q_1)$ where $Q_1$ is the first quartile and $Q_3$ is the third quartile. The number of clusters was decided based on the Dunn index, which is the ratio between the minimal inter-cluster distance to the maximal



intra-cluster distance. Fig. S2 shows the Dunn index as a function of the number of clusters, in which a decreasing trend is observed. The choice of using 7 clusters was made based on the competing need to have a high Dunn index and an ensemble of cluster centroids representing a variety of particle number size distribution shapes.

### 2.4   Modeling the seasonal and diurnal variability

Process based modelling was used to analyse to what extent the observed median diurnal particle number size distribution evolution at the surface were influenced by (i) aerosol dynamic processes (condensation and coagulation), (ii) dry deposition, (iii) below cloud scavenging, (iv) surface emissions of primary particles, (v) downward transport of aerosol particles into the CBL, (vi) dilution during the CBL development and (vii) transport of new particles from the FT into the CBL during convective downdrafts connected to strong precipitation events (Wang et al., 2016). ADCHEM (Aerosol Dynamics, gas and particle phase

CHEMistry and radiative transfer model (Hermansson et al., 2014; Öström et al., 2017; Roldin et al., 2011) was implemented as a stationary 1D column model for the measurement site. In the model, the lower atmosphere was divided into 50 vertical layers, each 50 m thick, extending from the surface up to 2500 m a.g.l. The vertical mixing was simulated with the first-order closure scheme (K-theory) described by Öström et al. (2017). Gas-particle partitioning of condensable vapours was not explicitly simulated. Instead, we used a particle size independent condensation growth rate (GR) as an unknown model fitting

parameter, keeping the growth rate constant throughout the vertical column. The below cloud scavenging of particles in the whole vertical model domain (0-2500 m) was modelled using the size-resolved precipitation scavenging parameterization from Laakso et al. (2003) as a function of the precipitation intensity.

As model input we used median dry and wet season diurnal cycles of meteorological parameters for the years 2008 and 2009, obtained from the Global Data Assimilation System (GDAS), downloaded from NOAA Air Resource Laboratory Real-

time Environmental Application and Display sYstem (READY) (Rolph, 2016). Median diurnal cycles of BL height, horizontal momentum flux, sensible heat flux, temperature, RH and horizontal wind speed vertical profiles are shown in Fig. S3 and S4. The probability of rainfall intensity ($f_i$) was obtained from observations at the K34 tower (Fig. S5) and grouped into 13 histogram bins representing rainfall with intensities from 0.1 to 30 mm h$^{-1}$. The effect of below cloud scavenging of particles in size bin $j$ was calculated using the probability distribution of rainfall intensity:

$$N_{j,t} = N_{j,t-\Delta t} \sum_{i=1}^{13} e^{-\beta_{i,j} f_i \Delta t} \qquad (2)$$

where $\Delta t$ is the model time step in seconds, $N_{j,t}$ and $N_{j,t-\Delta t}$ are the particle number concentrations in size bin $j$ at time $t$ and $t-\Delta t$, respectively, and $\beta_{i,j}$ is the precipitation scavenging rate coefficient (unit: s$^{-1}$) for rainfall with intensity $i$ and particles in size bin $j$. $f_i$ represents the probability of rainfall intensity. In this way, the modelled particle population is always affected by rainfall of different intensities, but to a much smaller extent ($f_i << 1$) than during a specific rain event with an

intensity $i$ for which $f_i = 1$.

In order to mimic the observed diurnal behaviour of particle number size distributions we included three different particle source types (Fig S6): (Type 1) an evening and night-time source of mainly Aitken and accumulation mode particles within



the surface layer, (Type 2) a source of accumulation mode particles entrained from the FT or from the RL into the CBL, and (Type 3) a source of mainly Aitken mode particles transported into the CBL with convective downdrafts following after intense rainfall events (Wang et al., 2016).

The downward vertical transport of particles into the BL was not modelled explicitly. Instead, this process was parameterized by adding an exchange rate of particles from the BL with particles from the FT (Eq. (3)):

$$\frac{dN_j}{dt} = \lambda f_{dwnd} (N_{dwnd,j} - N_j) \tag{3}$$

The term $\frac{dN_j}{dt}$ is the rate of change of the particle number concentration in size bin $j$, $\lambda$ is an estimated median BL air exchange rate due to convective downdrafts, $f_{dwnd}$ is the probability of convective downdraft which was estimated to be equal to the probability of precipitation with an intensity greater than 5 mm h$^{-1}$ occurring one hour earlier, and $N_{dwnd,j}$ is the concentration of particles in size bin $j$ in the downdraft. Eq. (3) was applied to all vertical model layers within the BL.

The model was initiated with the observed median particle number size distribution at noon for all model layers and was set to simulate the diurnal particle number size distribution evolution during 10 consecutive days with identical meteorological conditions and particle sources. The 10 days model period with identical diurnal conditions were used in order to assure that the modelled particle number size distributions reach a steady state, without any noticeable impact from the initial model conditions or variations from one day to another. Model fitted parameters, such as particle source strengths and condensation growth rates, were determined in order to reproduce the observed median diurnal trends in the particle number size distributions. In Sect. 3.2 we present results from the $10^{th}$ simulation day and compare the model results with the observed median particle properties during wet and dry seasons.

It should be mentioned that the model system cannot be fully constrained and different combination of parameter values can generate similar results. Therefore, the combination of parameter is not only based on the optimal agreement between model and observations, but also on the physical credibility of the parameter values. Choices were made in this way, for example, adopting lower growth rates (GR) during the night-time and during the dry season, when the condensation sink is substantially larger than during the wet season. The GR used in the model is lower than the median value of 5.5 nm h$^{-1}$ derived from the observed NPF events (Sect. 3.4). This is reasonable because NPF event days are typically characterized by higher growth rates (e.g., Hyvönen et al., 2005).

# 3    Results and discussion

## 3.1    Seasonal variability of submicrometer particle size spectra

Submicrometer particle number size spectra were measured between February 2008 and October 2014, consisting of 179644 valid samples, being 46% in the wet season (Jan to Jun) and 54% in the dry season (Jul to Dec), comprising 749 days with observations. Fig. 1 shows annual cycle in particle number concentrations observed between 2008 and 2014, clearly depicting the seasonal variability. During the wet season, pristine conditions prevailed, with median particle number concentration of





403 (196–1054) cm$^{-3}$ (percentiles 10 and 90 within parenthesis) (Table 1). These values are similar to previously reported observations for this region (∼400 cm$^{-3}$ on average, according to Martin et al., 2010a; Zhou et al., 2002). In the dry season, the site is affected by advection of biomass burning aerosols from Eastern and Southern Amazonia. Median particle number concentration reaches 1254 (557–2928) cm$^{-3}$ (Table 1). The values are compatible with previously reported values for dry

season from another site in the same forest reservation (ZF2) (averages in the range of 1080 to 1400 cm$^{-3}$, according to Rissler et al., 2004; Rizzo et al., 2010). In spite of the three fold increase from wet to dry season, observed particle number concentrations are well below the observations in the Amazonian state of Rondônia for example, a site heavily affected by biomass burning emissions where particle number concentration averages in the range of 5700 to 10440 cm$^{-3}$ (Artaxo et al., 2002; Brito et al., 2014; Rissler et al., 2006).

Figure 2 shows the median particle number size distribution for each season. One of the major differences between both seasons is the clear bimodal shape with pronounced Aitken mode (∼30 -100 nm) during wet season, compared against accumulation mode (∼100-600 nm) dominated dry season aerosol size distribution. The local minima between Aitken and accumulation modes observed in the wet season is known as the Hoppel minima (Hoppel et al., 1994), and it is an indicator of aerosol in cloud processing. The persistence of the Hoppel minima and the weak contribution of nucleation mode particles (diameter <

30 nm) during the wet season suggest that submicrometer aerosol sources from above the BL are relevant in this forest site.

To analyze dynamic variations of the particle number size distributions, one to three lognormal modes were fitted to each size spectra. Aitken and accumulation modes were almost ubiquitous in the wet season (92% and 100% of samples, respectively) (Table 1). In the dry season, the Aitken mode was less frequent (63%), likely being overwhelmed by the influence of accumulation mode particles from regional biomass burning. Geometric mean diameters ($d$) typically ranged between 50–85

nm for the Aitken mode and between 100–220 nm for the accumulation mode (Table 1). The nucleation mode was present in 73% of size spectra observed in the wet season, and in 38% of size spectra in the dry season, with mean geometric diameters in the range 10–25 nm (Table 1). This result is consistent with previous observations of decreased occurrence of nucleation mode from wet to dry season (Rissler et al., 2004, 2006; Rizzo et al., 2010; Zhou et al., 2002). A possible explanation is the increased aerosol loading during the dry season, acting as a sink for nucleation mode particles and for low-volatility organic compounds.

The median particle surface area, which is approximately proportional to the condensation sink of non-volatile vapours, was 24 $\mu m^2 cm^{-3}$ during the wet season and 102 $\mu m^2 cm^{-3}$ during the dry season.

### 3.2   Diurnal variability and modelling of processes that govern the behavior of submicrometer particle size spectra

Figure 3 shows median diurnal cycles of total ($N_{total}$) and modal ($N_i$) particle number concentrations observed during wet and dry seasons, as well as the variability of the geometric mean diameter ($D_{pg}$) of size distributions. The observations are

satisfactory reproduced by the model (Fig. 4). Table 2 summarizes the model fitted parameters in order to reproduce the observations, particularly the estimated source strength of particles Type 1 (particles emitted at the surface), Type 2 (particles entrained from above the CBL) and Type 3 (Aitken mode particles transported into the CBL with convective downdrafts). The diurnal variations in the contribution of the different particle source types to the particle number concentrations in the surface,





as well as the characteristic number size distribution for the particle source Type 1, 2 and 3 are shown in Fig. S6 and S7 in the supplementary material.

The overall diurnal behavior of $N_{total}$ and $D_{pg}$ did not show significant seasonal variation, in spite of substantially different particle number size distribution shape and total number concentration (Fig. 3a–d). Between 2:00 and 6:00 local time (LT) the

total particle number concentration decreases continuously, both during the wet and dry season (Fig. 3a–b). According to the model simulations, this is mainly attributed to dry deposition in a shallow NL (Fig. S3–4) and insignificant primary particle emissions from the surface. At the same time, the $D_{pg}$ is fairly stable (Fig. 3c–d), especially during the wet season. In the model, this combination of decreasing particle number concentrations and nearly constant $D_{pg}$ can only be achieved if the particle condensation growth rates are low.

Around 8:00 $N_{total}$ and $D_{pg}$ starts to increase rapidly during both seasons. A shift to larger particle sizes during daytime was also reported by Rissler et al. (2006). From Fig. 3 i–j, it is clear that the accumulation mode leads the diurnal particle number concentration increase. Nucleation mode also increased in the morning, but its contribution to total particle number concentration is smaller due to its relatively low frequency of occurrence and concentrations (Table 1). Part of this behavior, i.e., decrease of Aitken mode concentrations simultaneously to accumulation mode increase, could be explained by the growth

of Aitken into accumulation mode particles. The Aitken mean geometric diameter (not shown) increased in the morning (1 nm h$^{-1}$, on average), supporting this hypothesis. Particle growth can be driven by the condensation of low volatility vapors produced by photo-oxidation of biogenic volatile organic compounds (BVOC) (e.g., Chen et al., 2015; Claeys et al., 2004; Hu et al., 2015; Sá et al., 2017) and by in cloud processing (Hoppel et al., 1994; Krejci et al., 2005a). The model simulations confirm the influence of condensation growth to the observed morning behaviour, but also indicates the influence of another

relevant process: entrainment of accumulation mode particles (particle source Type 2), as the CBL develops between 7:00 and 10:00 LT. The origin of such accumulation mode particles could be both long-range transport and in cloud processed particles in the RL and in the lowermost FT. In the dry season, the entrainment of long-range transported particles into the surface level during the morning hours has also been documented in Western Amazonia (Brito et al., 2014). Model simulations require a greater Type 2 particle source strength in the dry season compared to the wet season (Fig. S6 and Table 2). Accordingly,

previous observations pictured the rainforest as a net fine mode particle sink (Ahlm et al., 2010; Rizzo et al., 2010), with higher deposition fluxes in the dry season.

In the afternoon (14:00–18:00), a decrease was observed in $N_{total}$ (Fig. 3a–b), $D_{pg}$ (Fig. 3c–d), and accumulation mode concentration (Fig. 3i–j). Heavy rain showers are frequent in the afternoon (Machado et al., 2004; Nunes et al., 2016 and Fig. S5). According to the model simulations, the decreasing total particle number concentration can partly be explained by below

cloud scavenging. However, scavenging alone cannot explain the observed decreasing $D_{pg}$ values, which was explained by the model simulations by the inclusion of a Type 3 particle source (Fig. S6 and S8). This particle source refers to convective downdrafts that transport FT air masses with low concentrations of accumulation mode particles but relatively high concentrations of Aitken mode particles into the CBL (Wang et al., 2016). In agreement with that, observations show an increase in the Aitken mode particle concentrations in the afternoon (Fig. 3g–h), reaching its highest values in the evening.



Around 17:00 LT the total particle number concentration starts to rise relatively steeply, led by increasing Aitken and nucleation mode concentrations (Fig. 3 e–h). At this time, the BL is becoming very shallow and surface RH increases from around 80% to 95% on average (Fig. S3–4). Surface emission of biological could explain the steeply increasing number concentration during evening, both during the wet and dry season. A possible mechanism is related to the rapid RH transition in the late

afternoon, which can trigger emission of submicrometer primary biological aerosol particles by rupturing and wet discharge of fungal spores. Fungal spores are mostly supermicrometer sized (e.g., Huffman et al., 2012), but their fragments and the liquid droplets discharged along with them are emitted in a broad range of sizes, from tenths of nm to $\mu$m (China et al., 2016; Elbert et al., 2007). In order to explain the small trends in $D_{pg}$ during the evening, model calculations indicate that the surface emitted particles (source Type 1, Fig. S6) need to be dominated by a mode around 65 nm in diameter, with a contribution of nucleation

mode particles only during the wet season.

Figure 4 shows modelled average particle number size distributions for the wet and dry season when one process at the time was turned off in the model. For the base case, the modelled particle number size distributions are almost identical with the observations during the wet and dry seasons, respectively. It is evident that the single process with largest influence on the modelled particle number size distribution, especially during the wet season, is the parameterized convective downdrafts. If the

model does not take these downdrafts into account, the size distributions both during the dry and wet season are completely dominated by a large accumulation mode with $D_{pg}$ > 200 nm. If we instead assume that the convective downdrafts contain completely particle free air, the model instead substantially underestimates the concentration of Aitken mode particles during the wet season. Thus, according to our model simulations, the convective downdrafts serve as an important source of new Aitken mode particles to the surface layer, especially during the wet season. However, precipitation, usually associated with

convective downdrafts, effectively decreases the concentrations of accumulation mode particles and cause a net reduction in the median total particle number and volume concentrations in the CBL in the afternoon (Fig. 3a–b). According to the model, condensation, i.e. secondary aerosol formation, is by far the largest source of submicron aerosol volume (mass), with estimated volume contribution of 0.33 and 0.43 $\mu m^3 cm^{-3}$ per day in the lowermost 1000 m of the atmosphere, during wet and dry seasons, respectively. Assuming a density of 1 g cm$^{-3}$, this corresponds to aerosol mass concentration values of 0.33 and

0.43 $\mu g\ m^{-3}$, representing 10-15% of typical fine mode aerosol concentrations in Amazonian forest sites (about 2.0 and 4.0 $\mu g\ m^{-3}$ in the wet and dry seasons, Martin et al., 2010b). The estimated particle mass yield by condensation also agrees with reported PM1 mass concentrations of SOA produced by gas-to-particle conversion pathways (about 0.3 $\mu g\ m^{-3}$ in the wet season, Chen et al., 2015).

### 3.3 Cluster analysis

Cluster analysis was applied to all particle number size distributions samples, normalized by particle number concentration. The number of clusters was chosen to be seven (Sect. 2.3.1). Figure5 shows the cluster centroids, as well as the frequency of occurrence in each season and along the entire dataset. Four of the clusters occurred mostly in the wet season, and three of the clusters predominately in the dry season. Figure 6 shows relative frequency plots associating cluster occurrence to particle




number concentration, particle light absorption coefficient, variations on equivalent potential temperature ($\Delta\theta_e$) and ambient RH.

The clusters predominant in the wet season (#1, #2, #3, #4) were associated with relatively clean conditions, with total particle number concentrations <500 cm$^{-3}$ most of the time (60–70% of samples in each cluster) and particle absorption coefficients less than 1 Mm$^{-1}$ (65-80% of samples in each cluster) (Fig. 6). On the other hand, the clusters predominant in the dry season (#5, #6, #7) occurred when particle number concentrations were relatively high (>500 cm$^{-3}$ in 65–80% of samples), as well as particle absorption coefficients (>1 Mm$^{-1}$ in 45–90% of samples) (Fig. 6ab). Rizzo et al. (2013) reported particle absorption coefficients of $1.0\pm1.4$ Mm$^{-1}$ for the wet season and $3.9\pm3.6$ Mm$^{-1}$ for the dry season, at the same sampling site.

Among the dry season clusters, cluster #6 occurred during intermediate aerosol loadings (Fig. 6ab), and may be associated with periods of seasonal transition. Accordingly, it shows some contribution from the Aitken mode (Fig. 5b), typical of the natural wet season particle size distributions, indicating that the contribution of biogenic aerosols, either from particle sources at surface or from above the BL, is present year round, consistent with the model simulations.

In the wet season clusters #1, #2, and #3 the Aitken mode is predominant, whereas in the dry season clusters #5 and #7 the accumulation mode dominates the particle size spectra (Fig. 5ab). Only one cluster showed a strong influence of nucleation mode (wet season cluster #2) (Fig. 5a). This is true even if the number of clusters increased until 20, reflecting the transient nature of the nucleation mode (Table 1). Clusters #4 and #6 had similar contributions from Aitken and accumulation modes, and its centroid holds the Hoppel minimum, suggesting the influence of in cloud processed secondary aerosol formation. The cluster centroids with a distinct contribution of the Aitken mode (clusters #1, #2, #3, #4 and #6) were mostly observed at periods of negative $\Delta\theta_e$ (Fig. 6c, 50–75% of samples in each cluster), i.e., periods with decreasing surface equivalent potential temperature, suggesting the influence of convective downdrafts. This result strengthens the hypothesis of a source of Aitken mode particles from above the BL, as suggested by model results from Sect. 3.2.

Cluster #1 was the only one that showed a relationship with both precipitation and downdraft events. Considering a lag time of $\pm 6$ hours, cluster #1 typically occurred after rainy periods (Fig. S9), simultaneously with negative $\Delta\theta_e$ (75% of samples with $\Delta\theta_e < 0$, Fig. 6c), and at periods with increasing particle number concentration. It suggests that the particle samples within this cluster originated from above the CBL. Accordingly, the cluster #1 centroid shows a predominance of particles in the lower range of Aitken mode, centered at 50 nm (Fig. 5a), similar to the size distribution used in the ADCHEM model simulations to represent the particle population in the convective downdrafts (Fig. S6).

Dry season cluster #7 had distinct features: it shows an abundance of particles in the upper part of accumulation mode (centered at 210 nm) (Fig. 5b); it occurred at relatively dry conditions (62% of samples occurred at RH<80%) (Fig. 6d); and it was associated with high particle light absorption coefficients (90% of samples had absorption coefficients > 1 Mm$^{-1}$) (Fig. 6b). Two case studies analyzed in the Supplementary Material suggest that the occurrence of cluster #7 is connected to the influence of fresh biomass burning aerosols (Fig. S11 and S12). Cluster #7 was predominant during events of intense biomass burning activity, and also when fire spots relatively close to the TT34 site were detected. Considering the whole time series, clusters #5 and #6 were more frequent than cluster #7 (Fig. 5d), indicating that aged biomass burning emissions is the major anthropogenic influence over the TT34 site in the dry season.



Cluster analysis was not able to associate a specific particle number size distribution shape when local wind came from SE, the direction of the city of Manaus, which is 60 km away. It suggests that Manaus urban plume influence is occasional, and that particles are processed on its way to the forest site, merging with the aerosol population above forest areas apparently without significant differences concerning to particle size spectra shape. Rizzo et al. (2013) identified 43 episodes of Manaus plume

influence over the TT34 tower site between 2008 and 2011, comprising 1.5% of their dataset.

### 3.4 New particle formation and particle burst events

Analyzing daily contour plots of particle number size distributions, the 749 days with observations were classified into the following categories: days with new particle formation and growth (NPF) events (comprising 3% of days); undefined event days (28%); non-event days (52%) and unclassified days (16%). The classification criterion was based on Dal Maso et al.

(2005), which states that NPF events have the following characteristics: i) A distinctly new mode of particles with diameters less than 25 nm must appear in the size distribution; ii) The mode must prevail over a time span of hours; iii) The new mode must show signs of growth. Events were classified as undefined when a burst of nucleation or Aitken mode particles was observed, prevailing for at least 1 hour, but without clear signs of growth (Fig. 7). Unclassified days were those with measurement coverage less than 12 hours or with intermittent sampling periods.

The observed frequency of NPF events is very low in comparison to other forest environments worldwide (e.g., Boy et al., 2008; Han et al., 2013; Held et al., 2004; Mäkelä et al., 2000). In a boreal forest site in Finland, which hold one of the longest time series of particle size distribution observations, NPF event days reach 24%, being most frequent during spring and autumn (Dal Maso et al., 2005). In this dataset, the percentage of days with events classified as undefined (28%) is similar to what has been reported in Finland (37%, according to Dal Maso et al., 2005).

Most NPF events started at daytime (Fig. S13) and during the wet season (19 out of 24 events). This is expected, since worldwide NPF event days are typically characterized by relatively low condensation sinks (e.g., Hyvönen et al., 2005). The median particle surface area, which is approximately proportional to the condensation sink of non-volatile vapours, was, on average, 18 $\mu m^2 cm^{-3}$ at the onset of NPF events. This can be compared with the total dataset median values of 24 and 102 $\mu m^2 cm^{-3}$, respectively during the wet and dry seasons (Table 1). Growth rates for the NPF events were calculated by fitting a

first order polynomial to the geometric mean diameter of nucleation and Aitken modes, following its growth between 10 and 50 nm. Median growth rate was 5.5 nm h$^{-1}$, ranging from 1 to 17 nm h$^{-1}$. For comparison, particle growth rates at a boreal forest in Finland during the summer typically range from 8 to 12 nm $^{-1}$ (Kulmala et al., 2004).

Undefined events were also most common in the wet season (161 out of 212 events), but, contrary to NPF events, the bursts of nucleation and Aitken mode particles occurred equally at daytime (46%) and nighttime (Fig. S13). It is possible that different

mechanisms result in the observed undefined events. Comparison of atmospheric conditions during diurnal NPF and undefined diurnal events (Fig. S14) show that particle surface area was typically higher in undefined events, while radiation and ozone mixing ratios were similar to NPF events. That leads to the interpretation of diurnal undefined events as failed NPF and growth events, like the ones observed in Finland (Mazon et al., 2009). Nocturnal undefined events coincide with the Aitken mode





particle number concentration increase observed in the evening (Fig. 3), associated with surface emissions, likely of biological particles. However, the mechanism behind the undefined events remain unclear.

NPF and growth events may be connected with the occurrence of convective downdrafts, since precipitation and negative $\Delta\theta_e$ were frequent 2 hours before such events (Fig. S14). Moreover, cluster #1 (Fig. 5a), associated with the transport of Aitken

mode particles from above the BL (Sect. 3.3), was characteristic of NPF events (Fig. S10). On the other hand, cluster #2 was uncommon during non-event days, and very frequent during both NPF and undefined events (Fig. S10), so that particle size distributions within this cluster could be used as tracers for both events.

## 3.5 Influence of precipitation and downdraft events on particle number size distributions

Along the particle size distribution measurement period, 86 wet season afternoon (12–16 LT) precipitation events with mod-

erate to strong intensities (>10 mm h$^{-1}$) were identified, typically lasting 1.5 hour. Constraining the analysis to the afternoon favors the selection of rain events from convective origin, in opposition to rain produced by squall lines. Decreases in surface equivalent potential temperature, i.e., negative $\Delta\theta_e$ values, were used as a proxy for the occurrence of convective downdrafts (Sect. 2.1). Previous investigations in Amazonia have shown the tranport of ozone to the surface in association with convective downdrafts (Betts et al., 2002; Gerken et al., 2016). Former airborne observations have shown strong vertical gradients of

particle number concentration above forest areas in Amazonia during the wet season (Krejci et al., 2005b; Wang et al., 2016), indicating a profusion of Aitken and nucleation mode particles in the FT, attributed to new particle formation in the outflow of convective clouds. Wang et al. (2016) showed case studies to attest the relevance of convective downdrafts as a source of particles in the Amazon during the wet season. Here, a similar analysis is presented, but based on a longer measurement time series. On one hand, a longer time series provides statistical significance, but, on the other hand, such analysis put together

events that occurred under different atmospheric conditions. It is important to mention that the method used here does not distinguish between deep and shallow convection and its different air mass exchange efficiencies, and does not consider the occasional presence of long range transported aerosol plumes in the FT at the onset of precipitation events.

Figure 8 shows a lag time analysis using the instant of maximum precipitation rate in each event as reference (zero lag). Average negative values of $\Delta\theta_e$ at the onset of rain events indicate the occurrence of downdrafts in most of the 86 selected

precipitation events. Simultaneously, a fast descrease in $H_2O$ mixing ratios was detected, suggesting the mixture of surface air with dryer air from the FT. The average effect of the selected afternoon precipitation events is an increase in the number concentration of small particles ($N_{<50}$, diameter less than 50 nm), and a decrease in the concentration of accumulation mode particles ($N_{>100}$), resulting on a net reduction in total particle number concentration (Fig. 8d–f). The combination of particle scavenging and a source of small particles above the BL is consistent with the afternoon behaviour of particle number size

distributions, discussed in Sect. 3.2 and Fig. 3. Analysing the relationship between particle concentrations and occurrence of downdrafts (Fig. S15), the trend of increased $N_{<50}$ particle concentrations with negative $\Delta\theta_e$, as reported by Wang et al. (2016), is confirmed, but is less clear. Particle scavenging ratios and the vertical transport of particles from above the BL needs further investigation and improved methods, taking into account the efficiency of air mass exchanges for different types of clouds and convection scales, as well as the presence of aerosol plumes above the BL.



## 4 Conclusions

This work reports a multi-year dataset of submicrometer particle size spectra in Amazonia. In the dry season, accumulation mode dominates the particle number size spectra, due to the continuous input of aged biomass burning particles from regional emissions. During the wet season, Aitken and accumulation modes, as well as the Hoppel minimum, are present, and the relative importance of the modes changes during the day. Cluster analysis, applied to normalized particle number size distributions, was useful to distinguish particle size spectra shape typical of the influence of fresh biomass burning aerosols, vertical transport of particles associated with convective downdrafts and nucleation events.

One of the major scientific questions supporting the continuous observation of particle number size distributions in Amazonia is: Was the lack of surface level NPF events observed during short-term experiments representative over all seasons? This dataset confirms that NPF and subsequent particle growth is rare in Amazonia, such that 24 events were detected in 749 days of observations (3%). This frequency is very low in comparison to other forest environments worldwide (e.g., Boy et al., 2008; Han et al., 2013; Held et al., 2004; Mäkelä et al., 2000). Several hypotheses arise to explain the scarcity of NPF and growth events in the Amazon: (i) concentration of inorganic precursors like $SO_2$ and $NH_3$ are relatively low, typically, <0.1 ppb (Andreae et al., 1990; Rizzo, unpublished dataset) and <0.8 ppb (Trebs et al., 2004) in the wet season; (ii) NPF inhibition by high emissions of isoprene (Kanawade et al., 2011; Kiendler-Scharr et al., 2009); (iii) NPF inhibition by high RH levels (Bonn and Moortgat, 2003; Hamed et al., 2011; Hyvönen et al., 2005); (iv) particle formation occurs at point sources, instead of regional sources, so that subsequent growth to the accumulation mode cannot be followed (Vana et al., 2008). Most of the few NPF events observed occurred after precipitation and simultaneously with decreasing surface $\theta_e$, linking NPF events to the occurrence of convective downdrafts. On the other hand, bursts of Aitken and/or nucleation mode particles without subsequent growth, named undefined events, were relatively frequent (28% of the measurement days). Undefined events did not show a clear diurnal pattern, nor a relationship with downdraft occurrence, so that the observed particle bursts may be a combination of different processes.

Since NPF events within the BL are infrequent in Amazonia, other particle sources must be active to support the observed submicrometer aerosol loadings. Analysis of observed and modelled diurnal patterns of particle size spectra indicates that, in order to maintain the particle number concentrations, the following particle sources are needed: (i) a source of Aitken mode particles within the BL at evening and night (Type 1); (ii) entrainment of accumulation mode particles in the morning, either from in cloud processing or long range transport (Type 2); and (iii) vertical transport of Aitken mode aerosols into the BL by convective downdrafts in the afternoon (Type 3). Modelling points to dry deposition and afternoon wet deposition as the major aerosol sinks within the BL during the wet season and coagulation during the dry season.

Results indicate that the transport of Aitken mode particles from convective downdrafts (Type 3) have large impact on the median particle number size distributions, especially during the wet season, in accordance with the mechanism proposed by Wang et al. (2016). Analysis of 86 afternoon precipitation events in the wet season confirm the input of small particles ($N_{<50}$) into the BL at the onset of strong rain events, but also indicate a net decrease in total particle number concentration due to scavenging, so that particle removal is not entirely compensated by the input of particles from downdrafts. To reproduce the



observed particle number concentrations after the typical afternoon rain events, model simulations indicate the existence of an evening and nighttime source of particles within the BL (Type 1). Since the measurement site is in a forest reserve where local pollution sources are not significant, this source is likely biogenic, either from primary emissions or secondary aerosol formation. Primary biological emissions could be trigged, for example, by fast RH variations in the late afternoon, through the

5      rupture and wet discharge of fungal spores (China et al., 2016; Elbert et al., 2007). The mechanisms behind nighttime surface particle emissions need further investigation.

*Data availability.*  The data sets used in this publication can be downloaded from http://lfa.if.usp.br/ftp/public/LFA_Processed_Data/.

*Competing interests.*  The authors declare that they have no conflict of interest.

*Acknowledgements.*  This work was supported by Fundação de Amparo à Pesquisa do Estado de São Paulo (FAPESP 08/58100-2; 13/05014-

10     0), Conselho Nacional de Desenvolvimento Científico (CNPq) and European Integrated FP6 project on Aerosol Cloud Climate and Air Quality Interactions (EUCAARI – 34684), under the scope of LBA experiment. We thank INPA (Instituto Nacional de Pesquisas da Amazônia) for the coordination work of the LBA Experiment and meteorological data. Financial support by the Academy of Finland Centre of Excellence program (project no 1118615), Swedish Research Council FORMAS (project no 2014-1445) and STINT (project BR2013-5210) are gratefully acknowledged. We thank key persons for the support on aerosol sampling and analysis: Alcides Ribeiro, Ana Lucia Loureiro,

15     Fernando Morais, Fábio Jorge, Glauber Cirino, Kenia T. Wiedemann, Lívia Oliveira and Simone Guimarães.



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





**Table 1.** Frequence of occurrence, median and percentiles 10 and 90 for particle number size distribution parameters during the wet and dry seasons. Total particle number concentration ($N_{total}$) and surface area were integrated from 10-600 nm particle size distributions. Particle concentration ($N$), geometric mean diameter ($d$) and geometric standard deviation ($\sigma$) are shown for the lognormal modes fitted to the size distribution samples: nucleation, Aitken and accumulation mode. Wet season period was defined as Jan–Jun, and dry season as Jul–Dec.

| | | Season | # of samples | occurrence | median | percent 10 | percent 90 |
|---|---|---|---|---|---|---|---|
| | $N_{total}$ (cm$^{-3}$) | Wet | | | 403 | 196 | 1054 |
| | Area ($\mu m^2 cm^{-3}$) | Wet | 82759 | | 24 | 7 | 64 |
| | $N_{total}$ (cm$^{-3}$) | Dry | | | 1254 | 557 | 2928 |
| | Area ($\mu m^2 cm^{-3}$) | Dry | 96885 | | 102 | 37 | 281 |
| Nucleation mode | N (cm$^{-3}$) | Wet | | | 40 | 10 | 121 |
| | d (nm) | Wet | 60527 | 73% | 25 | 18 | 25 |
| | $\sigma$ | Wet | | | 1.5 | 1.3 | 1.7 |
| | N (cm$^{-3}$) | Dry | | | 89 | 24 | 338 |
| | d (nm) | Dry | 37091 | 38% | 24 | 10 | 25 |
| | $\sigma$ | Dry | | | 1.6 | 1.3 | 1.7 |
| Aitken mode | N (cm$^{-3}$) | Wet | | | 181 | 80 | 540 |
| | d (nm) | Wet | 76182 | 92% | 67 | 48 | 81 |
| | $\sigma$ | Wet | | | 1.5 | 1.3 | 1.7 |
| | N (cm$^{-3}$) | Dry | | | 307 | 109 | 1159 |
| | d (nm) | Dry | 61465 | 63% | 71 | 52 | 85 |
| | $\sigma$ | Dry | | | 1.5 | 1.3 | 1.7 |
| Accumulation mode | N (cm$^{-3}$) | Wet | | | 159 | 40 | 480 |
| | d (nm) | Wet | 82759 | 100% | 172 | 103 | 209 |
| | $\sigma$ | Wet | | | 1.4 | 1.3 | 1.7 |
| | N (cm$^{-3}$) | Dry | | | 672 | 193 | 1766 |
| | d (nm) | Dry | 96885 | 100% | 161 | 104 | 217 |
| | $\sigma$ | Dry | | | 1.5 | 1.3 | 1.7 |





**Table 2.** Model fitted parameters and their estimated values for the wet and dry seasons. Fitted condensation growth rates are shown for daytime (6:00 to 18:00 LT) and nighttime (18:00 to 6:00 LT). The contribution of the different particle sources is given as an effective daily concentration contribution to the lowermost 1000 meters of the atmosphere.

| Parameter | Season | Estimated value |
|---|---|---|
| BL air change rate during convective downdrafts ($\lambda$) | Wet and Dry | 3 h$^{-1}$ |
| Condensation growth rate | Wet | daytime: 2 nm h$^{-1}$<br>nighttime: 0.2 nm h$^{-1}$ |
| | Dry | daytime: 0.5 nm h$^{-1}$<br>nighttime: 0.2 nm h$^{-1}$ |
| Secondary aerosol formation (condensation) | Wet | 0.333 $\mu m^3 cm^{-3} day^{-1}$ |
| | Dry | 0.432 $\mu m^3 cm^{-3} day^{-1}$ |
| Surface emissions of primary particles (Type 1, Fig S6) | Wet | 5.8 $cm^{-3} day^{-1}$ or 0.0023 $\mu m^3 cm^{-3} day^{-1}$ |
| | Dry | 9.3 $cm^{-3} day^{-1}$ or 0.0093 $\mu m^3 cm^{-3} day^{-1}$ |
| Entrainment of particles into the BL (Type 2, Fig S6) | Wet | 12.5 $cm^{-3} day-1$ or 0.024 $\mu m^3 cm^{-3} day^{-1}$ |
| | Dry | 37.5 $cm^{-3} day-1$ or 0.081 $\mu m^3 cm^{-3} day^{-1}$ |




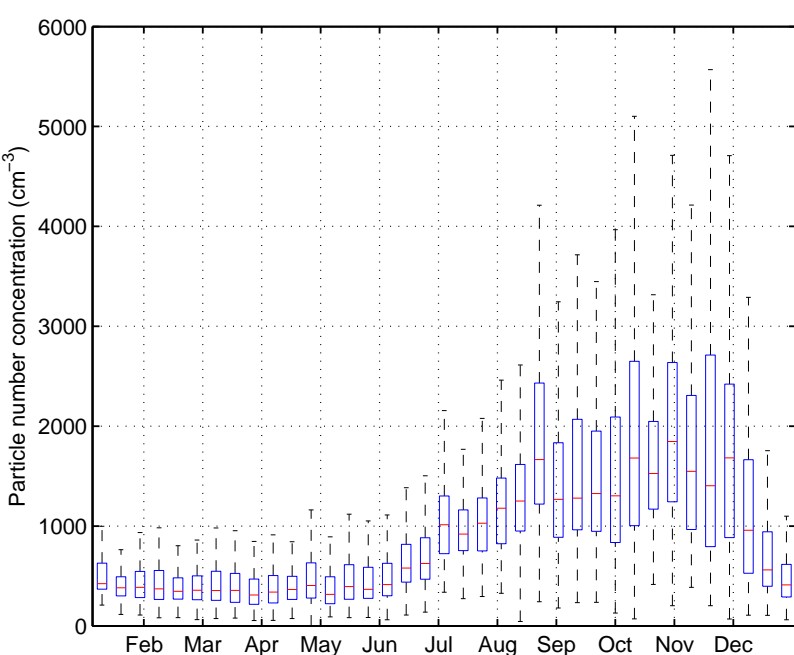

**Figure 1.** Statistics for a ten-day window for particle number concentrations between 2008 and 2014. The dataset is a combination of particle number concentration measurements and integrated particle size distribution measurements (10–600 nm). The red line represents the median; blue boxes extremes represent the 25 and 75 percentiles; black whiskers show the 1 and 99 percentiles.



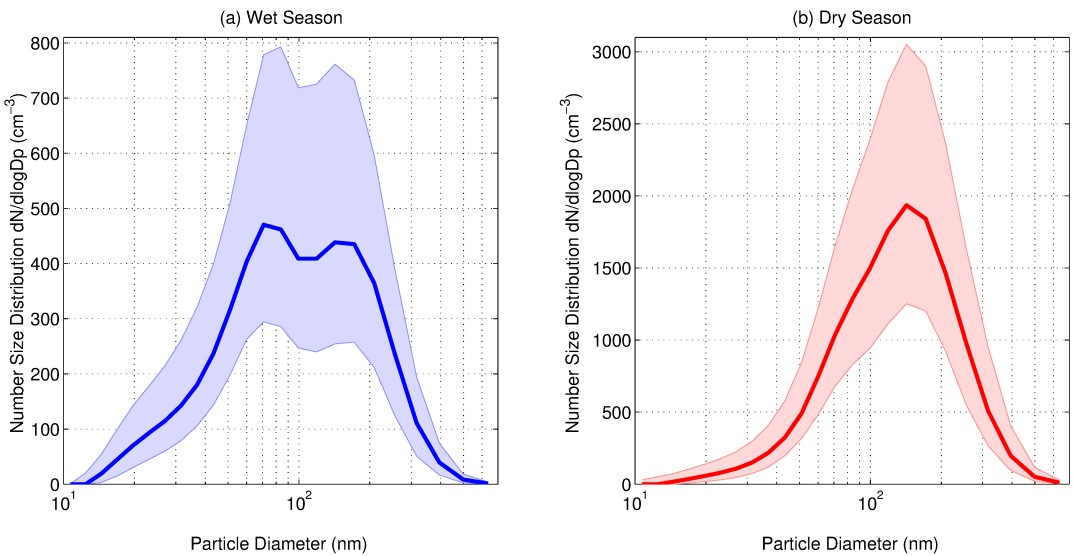

**Figure 2.** Median particle number size distributions for (a) wet and (b) dry season. Shadows represent the percentile range 25–75.



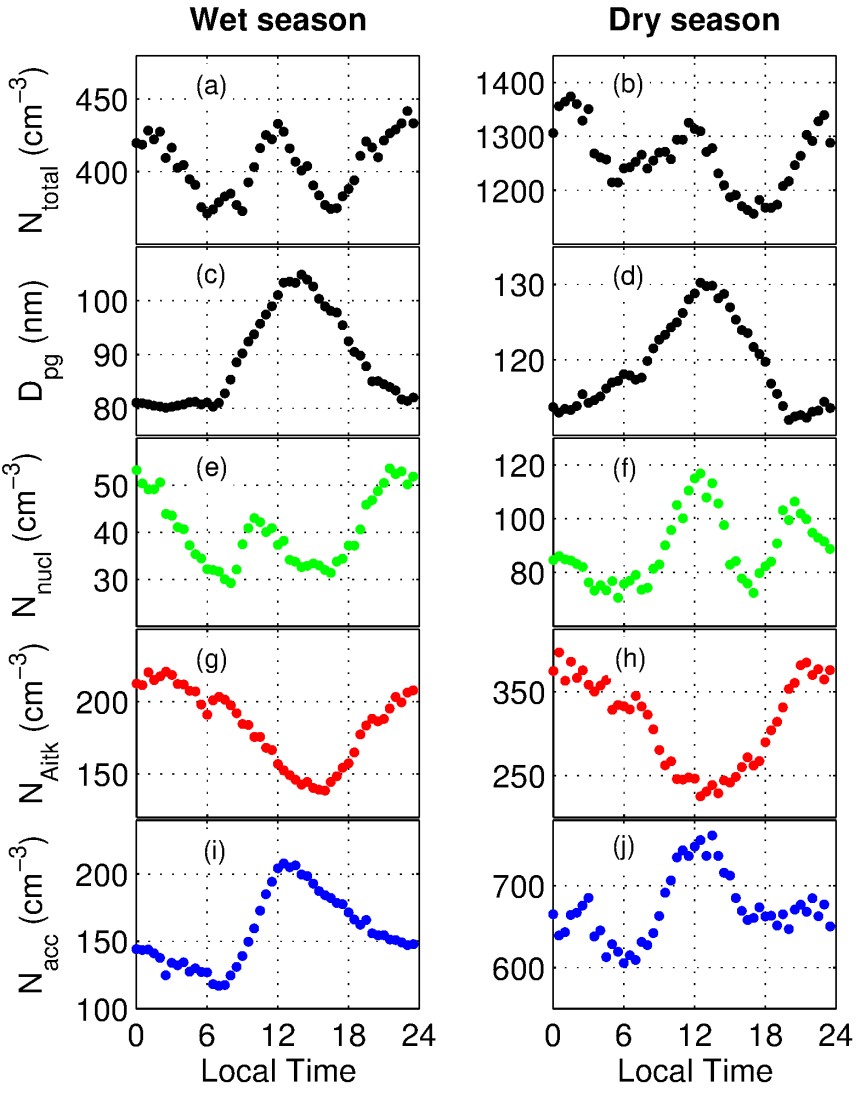

**Figure 3.** Median diurnal cycles for the wet and dry seasons. (a,b) total particle number concentration in the range 10–600 nm ($N_{total}$); (c,d) particle mean geometric diameter ($D_{pg}$); (e,f) concentration of the nucleation mode ($N_{nucl}$); (g,h), concentration of the Aitken mode ($N_{Aitk}$); (i,j) concentration of the accumulation mode ($N_{acc}$).





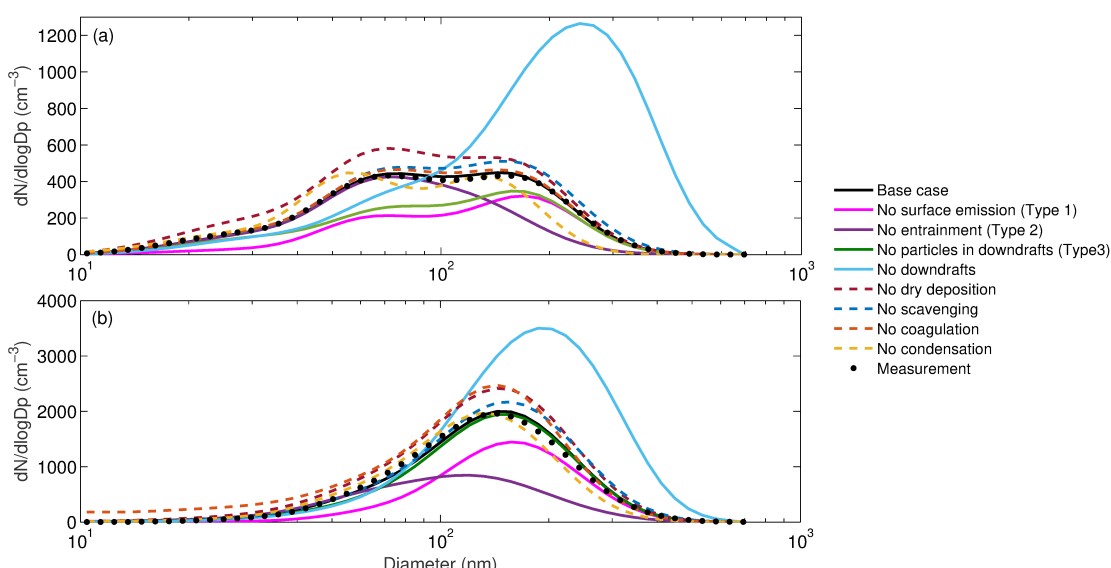

**Figure 4.** Modeled average particle number sized distributions for (a) the wet season and (b) the dry season. Shown are: the base case model runs, where all processes listed in Table 2 were considered; simulations without the Type 1 surface source of particles; simulations without the Type 2 source of particles entrained from above the BL; simulations with particle-free air in the convective downdrafts (Type 3); no convective downdrafts; no particle dry deposition losses; no below cloud scavenging; no coagulation; and no condensation. The measured median particle number size distributions are also displayed.




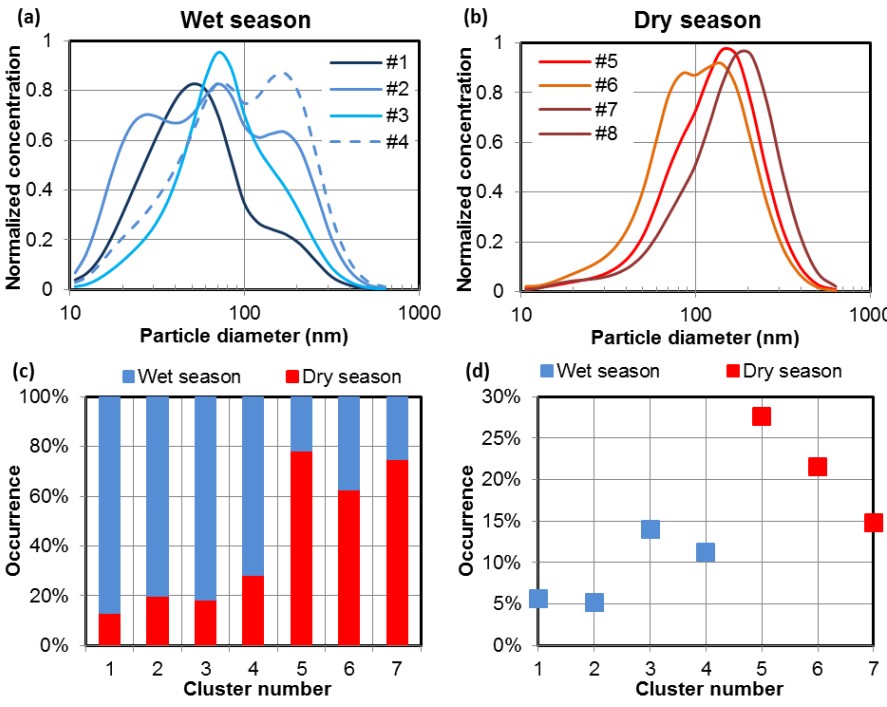

**Figure 5.** Normalized particle number size distribution cluster centroids occurring in the wet season (a) and dry season (b), percentage of occurrence of each cluster in each season (c) and within the entire dataset (d).





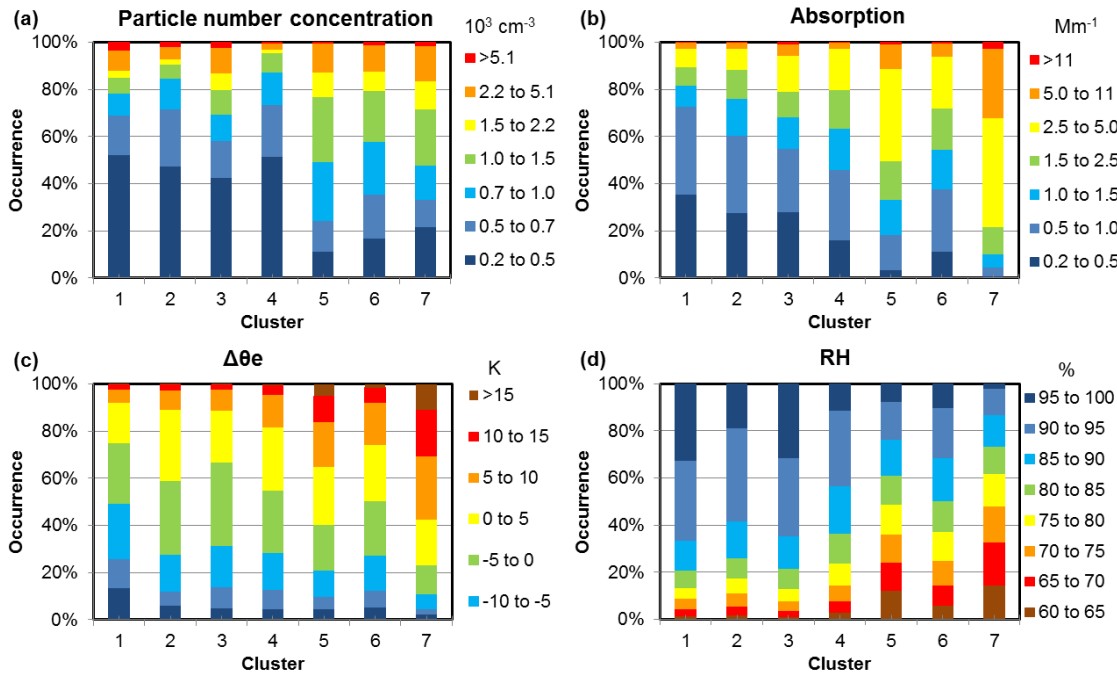

**Figure 6.** Stacked bar plots showing: (a) in which particle number concentration range each cluster was most frequent; (b) in which particle absorption coefficient range each cluster was most frequent; (c) in which equivalent potential temperature variation range ($\Delta\theta_e$) each cluster was most frequent; (d) in which RH range each cluster was most frequent. Wet season clusters (#1 to #4) showed aerosol number concentrations and absorption coefficients in background ranges, respectively 500 cm$^{-3}$ and 1 Mm$^{-1}$, contrary to dry season clusters (#5 to #7).





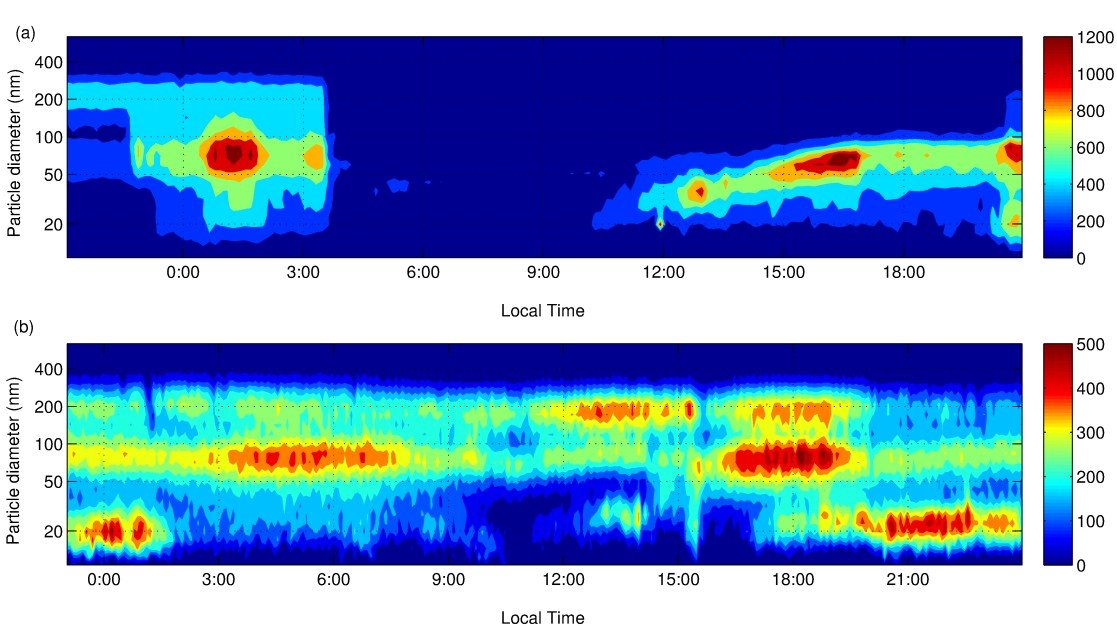

**Figure 7.** (a) Example of a new particle formation (NPF) and growth event starting on 21 Feb 21 10:00 LT. (b) Example of undefined events, on 04 Jun 2009. Particle bursts were observed at 0:00, 14:00 and 21:00 LT, without subsequent growth. In both plots, the color coding shows particle number concentration normalized by diameter channel, $dN/dlogD_p$, in cm$^{-3}$.



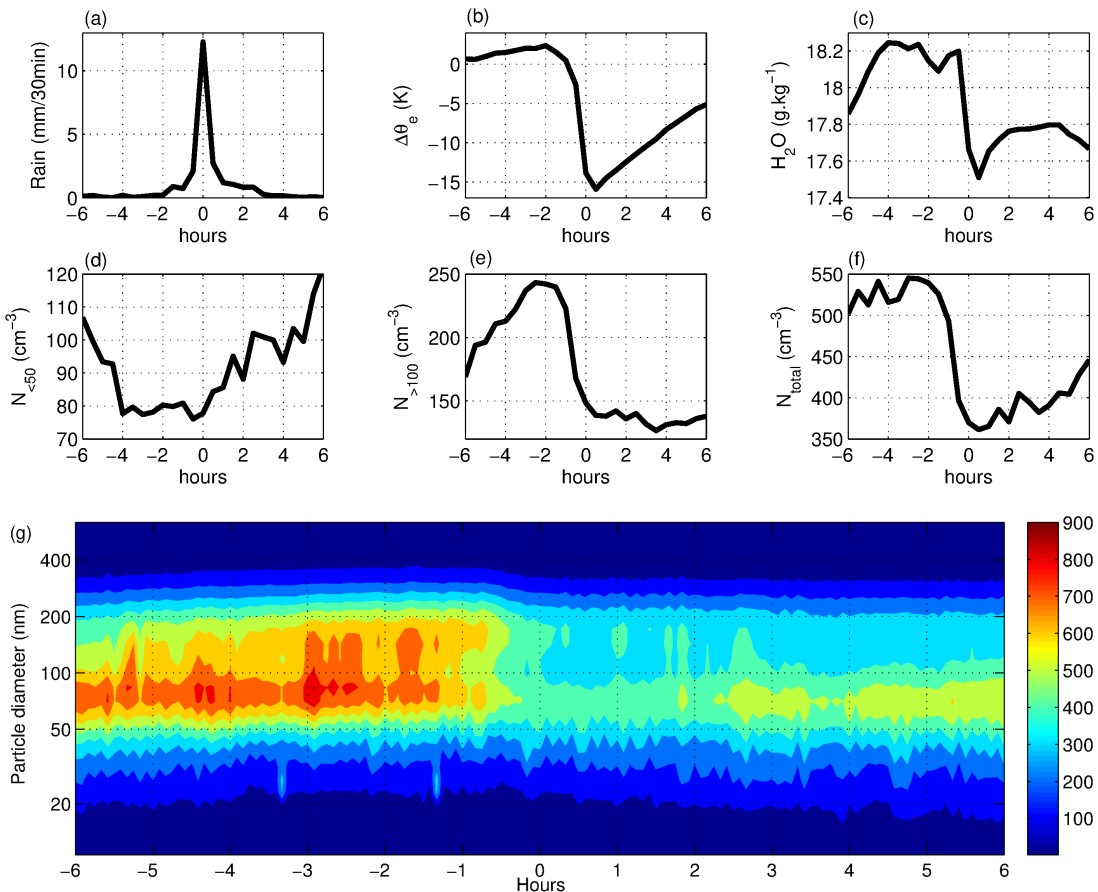

**Figure 8.** Mean values calculated within $\pm 6$ hours from the occurrence of strong precipitation events ($> 10$ mm h$^{-1}$). Zero time corresponds to the instant of maximum precipitation in each event. Only wet season afternoon events (12:00-16:00 LT) were considered in this plot. Top: (a) precipitation (accumulated mm in 30 min), (b) variations on equivalent potential temperature ($\Delta\theta_e$), (c) $H_2O$ mixing ratios. Middle: (d) number concentration of particles with diameter less than 50 nm ($N_{<50}$), (e) number concentration of particles with diameter greater than 100 nm ($N_{>100}$), (f) total particle number concentration ($N_{total}$). Bottom: (g) contour plot showing mean particle size distributions within $\pm 6$ hours of precipitation events. The color coding represents particle number concentration normalized by diameter channel, $dN/dlogD_p$, in cm$^{-3}$