# Peer review of "Multi-year statistical and modelling analysis of submicrometer aerosol number size distributions at a rain forest site in Amazonia"

_Atmospheric Chemistry and Physics, 2018_

## Referee Comment (RC1) · Anonymous Referee #1 · 8 Mar 2018

**Independent scientific review of "Multi-year statistical and modelling analysis of submicrometer aerosol number size distributions at a rain forest site in Amazonia" by L. V. Rizzo et al.**

**Major comments**

This paper discusses observations and features of ambient atmospheric aerosols in the Amazonia. Special focus is directed towards how the physicochemical properties is vastly changed from the wet season, with high occurrence of intensive rains, to the dry season. It is well known that the wet season are characterized by low particle number concentration and smaller size particles. While the dry season may show high abundance and concentration of biomass burning aerosols and an urban plume from the city of Manaus. In this paper, cluster analysis has been elegantly applied in order to explain many of the observed temporal features in aerosol size and aerosol mass concentration. Modelling trough the ADCHEM model has also been used in a comprehensive way in order to parameterize physical properties of importance for observed aerosol features. The main conclusion of the paper is the necessity of three stated aerosol sources in order to explain the observed physical properties of the aerosol.

The language is on a very high level and easy to follow. The methodology is technically sound and the inferred results and conclusions are well-supported by data from modelling, observations and references. The tables and figures are easy to read and understand. This paper deserves to be published after consideration of the minor comments stated below.

**Minor and specific comments**

Page 3, line 18-19. Please explain somewhere in the introduction why NPF is rare below BL in Amazonia but more common above the BL. This is unclear to me.

Page 6, line 27. State developer and version of the MATLAB software.

Page 6, line 29. Can these data-fences be illustrated somehow for increased understanding of the methodology?

Page 6, line 30. Explain the 1.5 factor please.

Page 7, line 13. It would be nice to get a very brief explanation of the "K-theory" here.

Page 11, line 3. "Surface emission of biological could…". Biological what? Particles?

Page 11, line 31. Should be "Figure 5".

Page 11, line 32 (Fig. 5). I do not understand the difference between "occurance in each season" and "occurrence along the entire dataset". In the figure both of these are seasonally divided.

Page 14, line 1. Are these biological particles primary or secondary?

Page 15, line 14. Please explain why high emissions of isoprene would inhibit NPF. Presence of BVOCs would enhance NPF?

Figure 1. Please explain the "ten-day window" in the figure caption.

Figure 5b. I can't recall that I ever heard anything about an 8th cluster as shown in fig. 5b?

Figure 5c-d. As mentioned earlier, I have some problem understanding the difference between 5c and 5d. Both are seasonally divided, still you claim that only fig. 5d show cluster occurrence for the entire dataset.

Figure 6b. Evidently, some absorption occur during the wet season. Can you elaborate in the text why we see this and what particle types that can be suspected for causing this absorption?

---

## Referee Comment (RC2) · Anonymous Referee #2 · 1 May 2018

This paper presents measurements and interpretation of the diurnal and seasonal cycle of aerosol at Cuieiras, Central Amazon. The paper is quite well written, and the discussions are relevant and point to new questions linking the atmospheric dynamics and the vertical profile of submicrometer size aerosol. I would point to one aspect that is the conclusion that free troposphere air has to be transported to the BL to explain the Aitken mode variability. In page 3, line 18, the authors mention that new particle formation NPF has been reported in the outflow level of deep convection. This is seen in high altitudes over 10 km. Convective downdrafts have their origin in the lower troposphere, below say 4 km. It is not clear how the particles would find their way from upper troposphere to lower troposphere to be available to be transported to the BL.

[Figure]

Perhaps the authors can comment on that. Page 3, last line – why do you say that SE of Cuieiras is the downwind direction, wouldn't it be the upwind direction? The data on $\Delta$ðİIJ∎E and RH that goes into the cluster analysis have what temporal frequency? Downdrafts are associated to sudden changes on the value of ðİIJ∎E so that the main drop is at the beginning of a downdraft event and no further decrease is observed in spite of the fac that ozone or aerosol may continue to be transported. It is not clear how this is taken into account when defining the variables that go into the cluster analysis. They should be simultaneous, right? Please clarify this point.

---

## Author Comment (AC1) · 19 Jun 2018

First, we would like to thank for the referee's comments, which helped to improve the manuscript. In the following, we will answer each specific point. Modifications in the text of the manuscript are also shown. Response to Referee #1 comments:

1. Page 3, line 18-19. Please explain somewhere in the introduction why NPF is rare below BL in Amazonia but more common above the BL. This is unclear to me.

The reason why NPF is rare below the BL in Amazonia is still not completely understood by the scientific community. In this paper, we document that NPF events in Ama-

zonia are indeed rare, and we affirm that supported by long term observations instead of the previously reported observations from intensive field experiments. To comply with the referee's comment, we decided to move a couple of sentences from the conclusion section (Page 15, line 10-17) to the introduction (Page 3, line 11), as follows: "Some hypotheses to explain the scarcity of NPF and subsequent growth events in the Amazon are: (i) concentration of inorganic precursors like SO2 and NH3 are relatively low, typically, <0.1 ppb (Andreae et al., 1990; Rizzo, unpublished dataset) and <0.8 ppb (Trebs et al., 2004) in the wet season; (ii) NPF inhibition by high emissions of isoprene (Kanawade et al., 2011; Kiendler-Scharr et al., 2009); (iii) NPF inhibition by high RH levels (Bonn and Moortgat, 2003; Hamed et al., 2011; Hyvönen et al., 2005); (iv) particle formation occurs at point sources, instead of regional sources, so that subsequent growth to the accumulation mode cannot be followed (Vana et al., 2008)." Please refer to comment #1 (reviewer 2) for the discussion about the NPF above the BL.

2. Page 6, line 27. State developer and version of the MATLAB software.

We added the required information, as follows: "Here, the MATLAB built-in function kmeans.m was used, available in the statistical package of version 2014a."

3. Page 6, line 29. Can these data-fences be illustrated somehow for increased understanding of the methodology? Page 6, line 30. Explain the 1.5 factor please.

These data-fences refer to the usual rule to label outliers. According to this definition, data points are considered outliers if they are smaller than f1 or larger than f3. The 1.5 factor is a choice, frequently the default option in boxplot algorithms, which corresponds to the whisker relative maximum length in a boxplot. Other factors could be used (e.g., Hoaglin et al., 1986), but 1.5 has been widely used for large datasets. If the data are normally distributed, the default 1.5 corresponds to a coverage of approximately 99.3% of the distribution. The boxplot in Figure 1 can be used to illustrate the outlier rule. We reformulated the sentence as follows: "Since this algorithm is very sensitive to outliers, these were omitted (0.13% of samples), using a standard rule for outlier labeling (e.g.,

[Figure]

Hoaglin et al., 1986). The inner fences (f1 and f3) of the data were calculated using the interquartile range (Q3 -Q1) as f1 = Q1-1.5(Q3-Q1) and f3 = Q3+1.5(Q3-Q1), where Q1 is the first quartile and Q3 is the third quartile. As an illustration, these data fences correspond to the extremes of the whiskers in Figure 1."

4. Page 7, line 13. It would be nice to get a very brief explanation of the "K-theory" here.

We included a brief explanation, as suggested by the reviewer: "The vertical mixing was simulated with the first-order closure scheme (K-theory) represented by a turbulent diffusion process. The vertical turbulent diffusion coefficient K was calculated based on a slightly modified Grisogono scheme (Jeričević et al., 2010), in which the vertical K-profile depends on the friction velocity and height of the planetary boundary layer (see Öström et al., 2017)."

5. Page 11, line 3. "Surface emission of biological could. . .". Biological what? Particles?

Indeed, the word "particles" was missing. We corrected that.

6. Page 11, line 31. Should be "Figure 5".

Ok, the typo was corrected.

7. Page 11, line 32 (Fig. 5). I do not understand the difference between "occurrence in each season" and "occurrence along the entire dataset". In the figure both of these are seasonally divided.

In Fig. 5c, we showed the percentage at which each cluster occurred in the wet and dry seasons. It is useful to explain that some clusters were predominant in the wet season, while others were more frequent in the dry season. On the other hand, in fig. 5d we showed the percentage of size distribution samples classified as cluster 1 to 7. To clarify that, we reformulated the caption in Fig. 5 and the text in page 11 as follows: "Figure 5. Normalized particle number size distribution cluster centroids occurring in

the wet season (a) and dry season (b), percentage of occurrence of each cluster in each season (c) and percentage of size distribution samples included in each cluster (d)." "Figure 5 shows the cluster centroids, as well as the frequency of occurrence in each season and the percentage of size distribution samples included in each cluster."

8. Page 14, line 1. Are these biological particles primary or secondary?

Based on the dataset we have, we cannot state whether these biological particles emitted at surface at night would be primary or secondary. We hypothesize that undefined particle burst events could be related to particle source Type 1 indicated by the model (Fig. S6), and indicate the rupturing of fungal spores as a possible mechanism for the nighttime emission of primary biological particles at surface (page 11, lines 4-8). We reformulated the sentence as follows: "Nocturnal undefined events coincide with the Aitken mode particle number concentration increase observed in the evening (Fig. 3), associated with surface emissions, likely of biogenic origin. Based on the available dataset, there are not enough evidences to affirm whether these particles would be primary or secondary, and the mechanism behind the undefined events remains unclear."

9. Page 15, line 14. Please explain why high emissions of isoprene would inhibit NPF. Presence of BVOCs would enhance NPF?

Observations at boreal forests have shown that NPF is more intense in the spring and fall compared to the summer, when isoprene emissions, dependent on light and temperature are higher (Kulmala et al., 2004). Observations at a mixed deciduous forest with large isoprene emissions during the summer also showed rare occurrence of NPF events, in spite of favorable conditions related to the presence of sulfuric acid and low condensation sink (Kanawade et al., 2011). A chamber experiment, in which NPF occurrence was monitored according to the amount of isoprene added over plant emissions, also indicated suppression of NPF at high isoprene concentrations (Kiendler-Scharr et al., 2009). More recently, Lee et al., (2016) indicated that NPF rarely occurs during summertime in isoprene dominated forests (ratio between isoprene and monoterpene concentrations greater than 1.0), even when monoterpene concentration was sufficient. Despite the observational evidences, the chemical mechanism for isoprene suppression of NPF is still unclear. Kiendler Scharr et al. (2009) suggested OH depletion by isoprene as the mechanism for NPF suppression, but this contradicts ambient observations at forests. Although the chemical mechanism is still unknown, NPF suppression by isoprene could explain in part the low frequency of NPF events observed in Amazonia, and this is the reason why we mentioned this possibility in the conclusion section. We included the most recent reference on this subject, Lee et al. (2016), in line 15 on page 15.

10. Figure 1. Please explain the "ten-day window" in the figure caption.

We reformulated the caption as follows: "Statistics for particle number concentrations considering observations between 2008 and 2014, calculated using a time window of 10 days (i.e., 1-10 Jan, 11-20 Jan, etc.). The dataset is a combination of particle number concentration measurements and integrated particle size distribution measurements (10–600 nm). The red line represents the median; blue boxes extremes represent the 25 and 75 percentiles; black whiskers show black whiskers extend to the most extreme data points not considered outliers."

11. Figure 5b. I can't recall that I ever heard anything about an 8th cluster as shown in fig. 5b?

Thank you for pointing out that. There is no 8th cluster, and we fixed this mistake in Figure 5b.

12. Figure 5c-d. As mentioned earlier, I have some problem understanding the difference between 5c and 5d. Both are seasonally divided, still you claim that only fig. 5d show cluster occurrence for the entire dataset.

Please refer to comment #7 (reviewer 1).

13. Figure 6b. Evidently, some absorption occur during the wet season. Can you elaborate in the text why we see this and what particle types that can be suspected for causing this absorption

To comply with the referee's comment, we included the following sentence in the first paragraph on Page 12: "In the dry season, the relatively large particle absorption coefficients are associated with the presence of biomass burning aerosols, while in the wet season absorption is attributed both to light absorbing biogenic aerosols and to long-range transported African biomass burning aerosols (e.g., Martin et. al, 2010b and references therein)."

Response to Referee #2 comments:

1. In page 3, line 18, the authors mention that new particle formation NPF has been reported in the outflow level of deep convection. This is seen in high altitudes over 10 km. Convective downdrafts have their origin in the lower troposphere, below say 4 km. It is not clear how the particles would find their way from upper troposphere to lower troposphere to be available to be transported to the BL. Perhaps the authors can comment on that.

We thank the reviewer for pointing out that, because it helped to improve the manuscript, delivering to the reader a better picture of the current scientific knowledge on this subject. What has been previously reported about aerosols above the boundary layer under clean conditions in Amazonia is: i) presence of nucleation mode particles in the upper troposphere (∼10 km), associated with the outflow of deep convective clouds (Andreae et al., 2018; Krejci et al., 2003), and ii) the presence of Aitken mode particles in the lower free troposphere (∼4-6 km) (e.g., Krejci et al., 2003; Wang et al., 2016). Andreae et al. (2018) proposed that the nucleation mode particles observed in the upper troposphere are formed by oxidation of biogenic VOCs brought up by deep convection, favored by an environment of low condensational sink and temperature, typical of the upper troposphere. Wang et al. (2016) hypothesize that the Aitken

mode particles observed in the lower free troposphere originated from the ultrafine particles observed at the upper troposphere, after condensational and coagulational growth. However, the means of transportation of particles from the upper troposphere to the lower free troposphere is still unclear, and have not been subject of the referred articles. On the other hand, the occurrence of convective downdrafts is a likely explanation for the transport of Aitken mode particles from the lower free troposphere to the boundary layer in Amazonia. Several authors reported a relationship between drops on surface equivalent potential temperature ($\theta$e), a proxy to the occurrence of convective downdrafts, with increased concentration of trace gases and particles at the surface (e.g., Betts et al., 2002; Gerken et al., 2016; Wang et al., 2016). Analysis of $\theta$e vertical profiles from radiosondes indicated the heights of 1500 to 2500 km as the probable level of origin of selected convective downdraft events in Amazonia, supposing that no mixing occurs between the descending air and the surrounding air (Betts et al., 2002; Schiro and Neelin, 2018). In the manuscript, we reformulated a couple of sentences in the introduction (page 3) to include part of this reasoning, as follows: "Another possible source is the downward transport of particles from above the BL, first suggested by Zhou et al. (2002). Observations reporting high concentrations of nucleation mode particles in the outflow of deep convective clouds in Amazonia ($\sim$10 km height) (Krejci et al., 2003, Andreae et al., 2018) and of Aitken mode particles in the lower free troposphere ($\sim$4-6 km height) (Krecji et al., 2003, Wang et al., 2016) reinforce this hypothesis. While the origin of the nucleation mode particles in the outflow of deep convection clouds could be attributed to new particle formation from the oxidation of uplifted biogenic VOCs, favored by low condensational sink and temperature typical of the upper troposphere (Andreae et al., 2018, Bianchi et al., 2016), the origin of the observed Aitken mode particles in the lower free troposphere is unclear. In-cloud processing (e.g., Hoppel et al., 1994; Lee et al., 2004) and particle formation through heterogeneous pathways (Jang et al., 2002; Lim et al., 2005; Limbeck et al., 2003) are possible mechanisms contributing to the presence of Aitken mode particles in the lower free troposphere. The downward transport of particles from the lower free

troposphere to the BL can occur either through convective downdrafts (Wang et al., 2016), typically observed in Amazonia (Machado et al., 2004; Nunes et al., 2016), and through continuous entrainment. Previous studies reported heights of 1500 to 2500 km as the probable level of origin of convective downdraft events in Amazonia (Betts et al., 2002; Schiro and Neelin, 2018). In this way, convective downdrafts could explain the transport of particles from the lower free troposphere into the BL, but the downward transport of particles from outflow regions of deep convective clouds is still uncertain, and requires more investigation."

2. Page 3, last line – why do you say that SE of Cuieiras is the downwind direction, would not it be the upwind direction?

Thanks for pointing out that. Indeed, E-SE is the upwind direction. We reformulated a group of sentences accordingly, to deliver the information in a more objective way: "The reservation is surrounded by primary tropical forest areas within a radius of about 500 km, except for the SE direction, in which Manus, a developing city with 1.8 million inhabitants (IBGE, 2015), sits 60 km away. Most of the time, the prevailing eastern trade winds blow over vast expanses of intact tropical forest before reaching the measurement tower (TT34) (2o35.6570' S, 60o12.5570' W, 110m a.s.l.). No biomass burning takes place in the reservation, but the site is affected by regional transport of biomass burning pollutants, especially in the dry season (July-December). Occasionally, the Manaus urban plume reaches the site (Rizzo et al., 2013). Measurements were carried out in two periods: from February 2008 to July 2010, and from November 2012 to October 2014."

3. The data on $\Delta\theta e$ and RH that goes into the cluster analysis have what temporal frequency? Downdrafts are associated to sudden changes on the value of $\Delta\theta e$ so that the main drop is at the beginning of a downdraft event and no further decrease is observed in spite of the fact that ozone or aerosol may continue to be transported. It is not clear how this is taken into account when defining the variables that go into the cluster analysis. They should be simultaneous, right? Please clarify this point.

Meteorological data were available at a temporal frequency of 30 minutes, while particle size distributions were available each 5-10 minutes. To associate the occurrence of particle size distribution clusters with $\Delta\theta$e and RH (Figure 6c-d), we used linear interpolation to match the time lines of meteorological and size distribution measurements. As a test, we also did the analysis using only the size distribution measurements concurrent to meteorological observations, obtaining similar results when compared to the linear interpolation procedure: positive $\Delta\theta$e values and lower RH values associated with the occurrence of the dry season clusters (#5-7). However, since it strongly reduced the number of data points, reducing the statistical strength of the analysis, we decided to show in Figure 6 the results obtained with the linear interpolation procedure. To clarify this point, we included a sentence on page 12, line 2, as follows: "Since meteorological data was available each 30 min, linear interpolation was used to match the time lines of meteorological observations and occurrence of particle size distribution clusters."

References:

[revised manuscript text omitted]